# Religious Diversity of Corporate Board and Firm Value: Evidence from South Korea

Wan-Yong Kim [1] and SungMan Yoon [2,*]

1 Department of Tax Accounting, Soongeui Women's College, Seoul 04628, Korea; wykim@sewc.ac.kr
2 Department of Business Administration, Seoul National University of Science and Technology, Seoul 01811, Korea
* Correspondence: ysm6123@seoultech.ac.kr

**Abstract:** When the board provides quality monitoring and advising to corporate managers, firms can achieve their goal, and so firm value increases. Board diversity is one of the issues that can affect the board effectiveness through influencing the quality of monitoring and advising. Hence, many prior studies have analyzed the effect of board diversity in various dimensions such as gender, age, race, ethnicity, education background on firm value or performance. However, prior studies do not reach out to the religious diversity of the board. So, this study investigates the relationship between religious diversity of the board and firm value. Using unique data of religion of directors for companies listed in Korea from 2008 to 2011, this study provides the following empirical results. In general, a high level of religious diversity has a positive impact on the firm value. However, when the degree of religious diversity of the board exceeds a certain point, religious diversity shows a negative relationship with the firm value. In addition, if the religion of directors is concentrated in one religion (when the level of diversity is low), firm value is lower than other control firms. The empirical finding of this study shows that religious diversity of the board matters in a similar way of other dimension of the board diversity.

**Keywords:** religiosity; religion; board diversity; board of directors; firm value; Tobin's Q

## 1. Introduction

Recently, diversity becomes one at the top of the corporate agenda. Companies have been making a steady effort to ensure that they welcome employees of all backgrounds. This is to attract and retainable employees regardless of gender, race, and ethnicity because diversity in terms of organization composition enhances inclusion of diverse perspectives and eventually induces corporate innovation. This diversity issue, however, is significant at not just employee level but also corporate director level.

The company's board of directors (hereafter, BOD) is one of the institutional mechanisms through which shareholders participate in corporate business. BOD has a vital role in the business; it plays a critical role in reducing agency problems by monitoring a management team (Jesen 1983); it also acts as an advisor to the management to maximize corporate profits (Fracassi and Tate 2012). Thus, it has a significant impact on business performance and firm value. So, how well the board does its role matters in maximizing firm value.

As aforementioned, the issue of board diversity, recently receives considerable attention as a way to improve the board effectiveness. The board with greater diversity tends to incorporate diverse perspectives, ideas, and information, which prevents the board from approving the management decision without checking and provides innovative consultation to the management. Hence, the board diversity in various dimensions enhances the quality of monitoring and advising.

Meanwhile, individual religious affiliation significantly impacts individuals' life values, perspectives, and attitudes. So, directors' religion likely has a significant influence

on their behavior, and so on the corporate management through BODs' monitoring and advising activities. Each religious group has its own value and belief distinct from other religious groups, particularly strictness and application of the value is quite different across different religious groups. Therefore, the board diversity in terms of religion may drive similar economic consequences with other dimensions of the board diversity. However, there is little research to examine the impact of the religious diversity of the board, due to perhaps data availability. Therefore, this study focuses on the religious diversity of directors and attempts to analyze how it affects firm value.

The remainders of this study are organized as follows. Section 2 provides literature review on board diversity, economic impact of religion and hypothesis development. Section 3 presents research model, variables measurement and sample selection. Section 4 shows the results of descriptive statistics, correlation, and regression analysis. Lastly, Section 5 summarizes results and provides conclusions.

## 2. Literature Review and Research Question

### 2.1. Literature on the Diversity of Board

The board has two roles: monitoring role and advisory role. More specifically, it has a responsibility to monitor the corporate management's self-serving behavior, which adversely affects firm value. Additionally, it should provide quality advice to the management related to business decisions: for example, when the management set corporate strategy and selects projects. Thus, when the board effectively does its role, firm performances increase since the management team cannot seek their private benefit at the shareholders' expense and can make better business decisions with optimal outcomes. So, effective monitoring and advising by the board eventually increase firm value.

Prior literature has studied what improves the effectiveness of the board from various dimensions: for example, board composition, director compensation, and director reputation. Particularly, the board composition has been extensively studied; board independence, the board size, board tenure, board busyness, politically connected board, expertise of board, background of directors, and so on. Of all these topics, board diversity is one of the main issues in the area of board composition research.

Then, why does board diversity matter to the board's effectiveness? First, according to social psychology studies, diversity fosters moderation in decision-making (Kogan and Wallach 1966). Homogeneous groups have homogeneity of views and incentives, which results in making unchecked decision and extreme outcomes (Kogan and Wallach 1966; Ferreira 2010; Bernile et al. 2018). Thus, the board with less diversity likely makes unchecked decisions than the board with greater diversity. Especially when their homogeneous view is similar to the management's view, it is less likely that the board with less diversity intensively monitors the management decision. Thus, the board effectiveness in terms of monitoring would deteriorate under the less-diversity board.

With respect to an advisory role, diversity affects creativity and innovation (Carter et al. 2003; Miller and Triana 2009). "Attitude, cognitive function, and beliefs are not randomly distributed in the population, but tend to vary systematically with demographic variables" (Robinson and Dechant 1997). Hence, heterogeneous groups in terms of demography have more diverse views and ideas (Milliken and Vollrath 1991). In addition, different demographic background indicates different information-processing way and network, which enhances a range of information of the board with greater diversity (Coleman 1988). Such a wide range of perspectives, ideas, and information likely improves the quality of the board's advice, and therefore promotes corporate innovation.

Based on the above argument, researchers have examined various dimensions in terms of board diversity such as gender, age, race, ethnicity, culture, education background. Bernile et al. (2018) report the lower firm risk of the board with greater diversity, using multiple dimensions of the diversity such as gender, age, ethnicity, expertise, education. Bennouri Moez et al. (2018) and Adams and Ferreira (2009) specifically focuses on gender

diversity, and shows the greater firm performance when firms have more gender-diverse directors on their board.

### 2.2. Literature on the Religion

Over the last several decades, there has been a significant attempt to incorporate religion in economics, finance, and accounting research. Especially, considering religion as an independent variable rather than a dependent variable, researchers have investigated whether and how religion influences corporate behavior, capital or product market, and economy.

When studying the economic consequences of religion, researchers mainly rely on the finding of positive correlation between individual religiosity and risk aversion; Miller and Hoffmann (1995) report that individuals with higher church attendance self-report themselves as risk-averse; Diaz (2000) shows a negative correlation between individual religiosity and the frequency of gambling. Since individual characteristics affect organization behavior, firms with more religious individuals are more likely to behave in a conservative way; firms headquartered in more religious cluster area show lower investment rate and less growth (Hilary and Hui 2009); firms with religious founders have lower leverage and less invest in fixed assets (Jiang et al. 2015). In terms of financial reporting, empirical findings also report the negative association between religiosity and financial reporting misstatement. They say religion induces managers not to accept unethical reporting practice. Hence, firms in religious areas exhibit lower incidences of financial reporting irregularities (McGuire et al. 2012); firms headquartered in more religious counties experience less stock crash since religious managers are not involved in bad-news-hoarding activities (Callen and Fang 2015).

Aforementioned studies, however, do not distinguish religious groups due to data availability. Most of these prior studies assume that all religious groups have similar behavioral norms, so individuals in every religious group show similar behavior. However, there are quite many religious groups around the world; Christians (Catholics, Protestants), Muslims, Hindus, Buddhists, Folk religionists, Jews, and so on. Each religious group has slightly different views on several issues. For example, Protestant churches have a very strong moral opposition to gambling and lotteries, while the Catholic church has tolerance toward a moderate level of gambling (Thompson 2001; Kumar et al. 2011). Based on this argument, Kumar et al. (2011) distinguish Catholics and Protestants and show that stock investors hold more lottery-type stocks in regions with a higher concentration of Catholics relative to Protestants.

### 2.3. Hypothesis Development: Relationship between Religious Diversity and Firm Value

Group diversity has many aspects (Williams and O'Reilly 1998). As aforementioned, prior literature on board diversity has examined various dimensions of the diversity such as gender, age, race, ethnicity, culture, expertise, board experience, and educational background. However, the religious diversity of the board has not received attention from practice and academia. We conjecture possible reasons for little attention as follows.

Not all aspects of the board diversity necessarily matter for corporate policies. One argues that religion's impact on individuals is likely to be similar to the impact of culture, so separating the religion effect and the cultural effect does not have additional implications. La Porta et al. (1999) indeed use religion as a proxy for culture in their study. On the other hand, there may be an argument that religious groups have a similar behavioral norm. So, which religious groups individuals belong to does not matter. Rather, whether individuals belong to any religious group matters. Under this argument, not the religious diversity but the degree of religiosity is the issue in the corporate governance.

Despite the possible explanation for little implication of religious diversity, the religious diversity of the board would affect the board effectiveness and firm value in both positive and negative ways. Even if each religious group's overall norm and value seem to be similar, its strictness or application is quite different. So, the attitude of individuals

can be different based on religious affiliation. Thus, it is a faulty generalization to put all religious groups in one category and assume they have similar attitudes and behaviors. Kumar et al. (2011) provide empirical evidence that individuals of each religious groups show different behavior in terms of investment in stock market. Barsky et al. (1997) also show that Catholics are less risk averse than Protestants, and more tolerate of speculative risk taking. Thus, similar to other dimensions of the diversity, the religious diversity likely implies various attitudes and perspectives. So, the diverse religious affiliation in the board brings more views, ideas, and information, which consequently enhances quality of advice. In addition, such diversity in terms of religious affiliation prevents the board from being a homogeneous group and avoids unchecked decision-making, which likely results in a high-quality monitoring. Therefore, the religious diversity of the board is likely to affect the firm value positively.

On the other hand, a negative impact could also be expected. Diversity would generate more conflicts and make it harder to arrive at a consensus, which disrupts the decision-making process (Adams and Ferreira 2009). Particularly, religious individuals are likely to adhere to their own views and agenda when conflict arises. Hence, the religious diversity of the board delays the corporate decision-making process and consequently reduces firm value. Thus, the hypothesis is stated in a null form.

**Hypothesis 1 (H1).** *Religious diversity of the board does not affect firm value.*

### 3. Research Methodology and Data

Data on the religion of corporate BOD members are hard to obtain since there is no disclosure rule mandating corporate disclosure of directors' religion. Furthermore, companies do not even voluntarily disclose directors' religion since it is private information. Thus, due to data availability, studies on the relationship between religious factors and firm values have not been actively examined so far. However, Korea Securities Dealers Automated Quotation(hereafter, KOSDAQ) Association conducts a survey for four years from 2008 to 2011 on directors' religion of manufacturing companies listed in the KOSDAQ stock market. We obtain the data of directors' religion from KOSDAQ Association and so limit our sample to KOSDAQ-listed companies from 2008 to 2011. Admitting that our sample is limited sample, we utilize this data to examine our research question because this data is unique and rare. In addition, we obtain financial data for Tobin's Q and various control variables from KIS-VALUE database which is one of three corporate data providers in Korea. Our final sample consists of 1092 firm-year observations.

Our research model is presented below in Equations (1) and (2). The dependent variable is Tobin's Q, which represents the firm value. The main variable of interest is Diversity, the religious diversity of directors. We measure the religious diversity as how many different religions board directors belong to. More specifically, Diversity variable is defined as the number of religions of directors divided by 7. We use 7 as a denominator because the religions of the directors in our sample consist of 7 religions. If $\beta1$ in Equation (1) is positive (or negative), it means that the religious diversity of corporate board members increases (or decreases) the firm value.

It is also possible that the religious diversity enhances the firm value up to a certain level of religious diversity, but deteriorates the firm value after exceeding that level. Hence, in Equation (2), the $Diversity^2$ variable is added to investigate whether a relationship between Diversity and Tobin's Q has a inverse U shape rather than a linear relationship. If $\beta1$ in Equation (2) is positive and $\beta2$ is negative, it means that the positive relationship is reversed after the religious diversity exceeds a certain level. We control various factors that affect Tobin's Q: firm size (SIZE), debt ratio (Leverage), foreign shareholders' equity (Foreign), R&D expenses (RnD), total asset growth rate (Growth), and firm age (AGE) and SME dummy variables (SME). We also include year and industry dummy variables to control for year and industry fixed effects. We winsorize the top and bottom 1% of the variables to remove outlier effect.

$$\text{Tobin's Q}_{i,t} = \alpha_0 + \beta_1 \text{Diversity}_{i,t} + \beta_2 \text{SIZE}_{i,t} + \beta_3 \text{Leverage}_{i,t} + \beta_4 \text{Foreign}_{i,t} + \beta_5 \text{RnD}_{i,t} + \beta_6 \text{Growth}_{i,t} + B_7 \text{AGE}_{i,t}$$
$$+ \beta_8 \text{SME}_{i,t} + \beta\sum\text{YEAR}_{i,t} + \beta\sum\text{IND}_{i,t} + \varepsilon_{i,t} \tag{1}$$

$$\text{Tobin's Q}_{i,t} = \alpha_0 + \beta_1 \text{Diversity}_{i,t} + \beta_2 \text{Diversity}^2_{i,t} + \beta_3 \text{SIZE}_{i,t} + \beta_4 \text{Leverage}_{i,t} + \beta_5 \text{Foreign}_{i,t} + \beta_6 \text{RnD}_{i,t}$$
$$+ \beta_7 \text{Growth}_{i,t} + \beta_8 \text{AGE}_{i,t} + \beta_9 \text{SME}_{i,t} + \beta\sum\text{YEAR}_{i,t} + \beta\sum\text{IND}_{i,t} + \varepsilon_{i,t} \tag{2}$$

Tobin's Q: Tobin's Q is measured as the sum of market value of common stock, preferred stock, and book value of debt divided by book value assets.
Diversity: the religious diversity (# of religion types of directors divided by 7)
SME: Indicator variable that equals i if the firm is small and medium enterprise, and 0 otherwise.
SIZE: the natural logarithm of total assets.
Leverage: the sum of long-term debts and short-term debts divided by total asset.
Foreign: the ownership percentage of foreign investors.
RnD: the ratio of research and development expense divided by total sales.
Growth: sales growth.
AGE: The natural logarithm of the firm age.

## 4. Results of Analysis and Discussion

Table 1 shows descriptive statistics of variables. The average of Tobin's Q is 1.2231, in the range of 0.472 to 5.3363, and the average of diversity is 0.714, in the range of 0.1672 to 1. In particular, it is peculiar that directors have seven major religions in the top 25% or more of the diversity variable. In other words, companies in which BOD members' religions are dispersed into 7 religions have a high level of religious diversity.

**Table 1.** Descriptive Statistics (N = 1178).

| Variable | Mean | Std. Dev | Min | 25% | Median | 75% | Max |
| --- | --- | --- | --- | --- | --- | --- | --- |
| Tobin's Q | 1.2231 | 0.7381 | 0.4720 | 0.7982 | 1.0141 | 1.3850 | 5.3363 |
| Diversity | 0.7140 | 0.2830 | 0.1672 | 0.5200 | 0.6672 | 1 | 1 |
| SIZE | 25.1411 | 0.7792 | 23.5223 | 24.6471 | 25.0751 | 25.5794 | 27.2420 |
| Leverage | 0.3940 | 0.1900 | 0.0512 | 0.2432 | 0.3881 | 0.5330 | 0.8572 |
| Foreign | 3.2124 | 6.1332 | 0.0100 | 0.1320 | 0.5610 | 2.9500 | 30.6910 |
| RnD | 0.0100 | 0.0221 | 0 | 0 | 0 | 0.0110 | 0.1192 |
| Growth | 20.6340 | 55.8651 | −59.2221 | −4.3431 | 10.0130 | 28.7310 | 355.1920 |
| AGE | 2.9622 | 0.5091 | 1.3862 | 2.5654 | 2.9444 | 3.3324 | 4.0071 |
| SME | 0.5400 | 0.4990 | 0 | 0 | 1 | 1 | 1 |

*Note*: The definitions of variables are the same as the measurement of variables in Equation (1).

Table 2 shows the Pearson correlation between variables. Correlation between the religious diversity and Tobin's Q is significantly negative at the 5% level, which means that the higher the level of religious diversity, the lower the firm value. In addition, among the control variables, Size, Foreign, RnD and AGE show significant correlation with Tobin's Q at 1% level, respectively, and Growth and SME also have significant correlation at 5% and 10% level, respectively. This correlation result indicates that high foreign ownership, high level of R&D investment, high asset growth, low company age, small and median size is correlated with high firm value.

**Table 2.** Pearson's Correlation of Variables.

| | Tobin's Q | Diversity | SIZE | Leverage | Foreign | RnD | Growth | AGE |
|---|---|---|---|---|---|---|---|---|
| Diversity | −0.0572 ** (0.0498) | 1 | | | | | | |
| SIZE | −0.0992 *** (0.0000) | −0.0645 ** (0.0267) | 1 | | | | | |
| Leverage | −0.0362 (0.2143) | 0.0402 (0.1685) | 0.2712 *** (0.0000) | 1 | | | | |
| Foreign | 0.1405 *** (0.000) | −0.0031 (0.9164) | 0.2579 *** (0.0000) | −0.0560 * (0.0547) | 1 | | | |
| RnD | 0.1618 *** (0.0001) | −0.0114 (0.6950) | −0.0799 *** (0.0061) | −0.0542 * (0.0628) | −0.0504 * (0.0840) | 1 | | |
| Growth | 0.0694 ** (0.0172) | 0.0022 (0.9392) | 0.0248 (0.3949) | 0.0726 ** (0.0128) | −0.0001 (0.9973) | 0.0446 (0.1261) | 1 | |
| AGE | −0.1527 *** (0.0001) | −0.0906 *** (0.0002) | 0.2059 *** (0.0000) | 0.0589 ** (0.0432) | −0.0435 (0.1355) | −0.1271 *** (0.0000) | −0.1024 *** (0.0004) | 1 |
| SME | 0.0526 * (0.0713) | 0.0539 * (0.0644) | −0.5840 *** (0.0000) | −0.1783 *** (0.0000) | −0.1998 *** (0.0000) | 0.0977 *** (0.0008) | −0.0289 (0.3225) | −0.0885 *** (0.0024) |

*Note1*: *, **, and *** indicate significance at the 10%, 5%, and 1% levels, respectively. *Note2*: The definitions of variables are the same as the measurement of variables in Equation (1).

Table 3 shows the results of OLS regression that analyzed the effect of religious diversity on firm value. In the results of Model 1, Diversity shows a statistically significant coefficient of 0.1399 ($p < 0.1$), which shows that firms with high religious diversity have high firm value. In the results of Model 2, Diversity also shows a statistically significant positive coefficient of 1.3008 ($p < 0.05$), while Diversity$^2$ shows a negative coefficient −1.061 ($p < 0.01$). This Model 2 result indicates that the religious diversity of board directors can improve firm value through providing various views and ideas to management decision-making, but if religion is too diverse, it deteriorates firm value due to more conflict and delay of consensus.

Among the control variables, in both Model 1 and Model 2, SIZE, Leverage, Foreign, RnD, and Growth show a statistically significant coefficient. It means that firms with smaller size, lower debt ratio, high foreign ownership, high R&D investment, and high growth have high firm value.

In Table 4, we additionally use fixed and random effect models to address endogeneity problems and measurement errors. We conduct this additional analysis only for Model 2 since Table 3, results of more sophisticated empirical model, confirms inverse-relationship between the religion diversity and firm value. In both the fixed effect model and the random effect model, the coefficients of Diversity is 1.7171 ($p < 0.05$) and 1.3985 ($p < 0.05$) at statistically significant levels, respectively. In particular, Diversity$^2$ also still shows statistically significant and negative coefficients in both the fixed effect and random effect model. These results provide strong evidence that the religious diversity does not have an unconditional positive impact on the firm value, but positively affect the firm value only up to an appropriate level. These results can be said to be more powerful statistically than Table 3.

So far, we suggest that low religion diversity generally reduces firm value. However, firms with a one-religion board which means all board members have the same religion, an extreme case of low religion diversity, could do more social response activities and ethical management behavior, which may result in improved firm value. Thus, we also test whether one-religion board affects firm value differently from other firms with low religion diversity. For this analysis, we include One Religion instead of Diversity in the Equation (1).

**Table 3.** OLS Regression Results.

| Variable | Model 1 | | Model 2 | |
|---|---|---|---|---|
| | Coefficient | *t*-Stat. | Coefficient | *t*-Stat. |
| Diversity | 0.1399 * | 1.84 | 1.3008 ** | 2.36 |
| Diversity$^2$ | - | - | −1.0610 *** | −2.64 |
| SIZE | −0.0798 ** | −2.17 | −0.0795 ** | −2.17 |
| Leverage | −0.0130 * | −1.85 | −0.0132 * | −1.88 |
| Foreign | 0.0191 *** | 6.10 | 0.0196 *** | 6.20 |
| RnD | 1.0908 ** | 2.03 | 0.9926 * | 1.84 |
| Growth | 0.0008 ** | 2.27 | 0.0008 ** | 2.14 |
| AGE | −0.0574 | −1.31 | −0.0696 | −1.59 |
| SME | 0.2131 | 0.40 | 0.0195 | 0.36 |
| _cons | 2.5857 *** | 2.76 | 2.2166 ** | 2.35 |
| ∑Industry | Included | | Included | |
| ∑Year | Included | | Included | |
| F-stat. | 5.16 *** | | 5.23 *** | |
| Adj. R$^2$ | 0.1409 | | 0.1458 | |
| Observations | | 1178 | | |

*Note1*: *, **, and *** indicate significance at the 10%, 5%, and 1% levels, respectively. *Note2*: The definitions of variables are the same as the measurement of variables in Equation (1).

**Table 4.** Fixed and Random Effect Model Regression Results.

| Variable | Fixed Effect Model | | Random Effect Model | |
|---|---|---|---|---|
| | Coefficient | *t*-Stat. | Coefficient | *t*-Stat. |
| Diversity | 1.7171 ** | 2.39 | 1.3985 ** | 2.51 |
| Diversity$^2$ | −1.3308 *** | −2.61 | −1.1562 *** | −2.88 |
| SIZE | −0.1750 ** | −2.22 | −0.0279 | −0.66 |
| Leverage | 0.0005 | 0.05 | −0.0088 | −1.41 |
| Foreign | 0.0089 | 1.49 | 0.0130 *** | 3.50 |
| RnD | −0.9563 | −1.40 | 0.1890 | 0.35 |
| Growth | 0.0009 *** | 3.02 | 0.0008 *** | 2.97 |
| AGE | 0.6571 *** | 3.28 | −0.0364 | −0.66 |
| SME | — | — | 0.1366 * | 1.87 |
| _cons | −5.9470 *** | −3.40 | −0.2347 | −0.22 |
| SME FE | Included | | - | |
| Industry FE/RE | Included | | Included | |
| Year FE/RE | Included | | Included | |
| R$^2$ | 0.0764 | | 0.0524 | |
| Observations | | 1178 | | |

*Note1*: *, **, and *** indicate significance at the 10%, 5%, and 1% levels, respectively. *Note2*: The definitions of variables are the same as the measurement of variables in Equation (1).

In Table 5, OLS regression results show that One Religion has a statistically significant coefficient of −0.0115 ($p < 0.01$). It indicates that firm value is lower in firms with one-religion board than in other firms. Results of fixed or random effect models also show the

same results: statistically significant coefficient of −0.1203 ($p < 0.05$) and −0.1451 (p < 0.01), respectively. This result declines our argument that the religious diversity of a one-religion board works differently compared to other low-religion-diversity firms.

**Table 5.** Additional Test (one religion concentration).

| Variable | OLS Regression | | Fixed Effect Model | | Random Effect Model | |
|---|---|---|---|---|---|---|
| | Coefficient | *t*-Stat. | Coefficient | *t*-Stat. | Coefficient | *t*-Stat. |
| One Religion | −0.0115 *** | −2.66 | −0.1203 ** | −2.07 | −0.1451 *** | −3.21 |
| SIZE | −0.0803 ** | −2.19 | −0.1829 ** | −2.32 | −0.0267 | −0.63 |
| Leverage | −0.0131 * | −1.86 | 0.0013 | 0.15 | −0.0088 | −1.40 |
| Foreign | 0.0194 *** | 6.13 | 0.0092 | 1.54 | 0.0131 *** | 3.51 |
| RnD | 1.0398 * | 1.93 | −1.0414 | −1.53 | 0.1978 | 0.37 |
| Growth | 0.0008 ** | 2.21 | 0.0009 *** | 3.00 | 0.0008 *** | 2.99 |
| AGE | −0.0649 | −1.48 | 0.6400 *** | 3.19 | −0.0401 | −0.73 |
| SME | 0.0208 | 0.39 | - | - | 0.1389 * | 1.91 |
| _cons | 2.5774 *** | 2.76 | −5.5992 *** | −3.20 | 0.1856 | 0.18 |
| SME FE | - | | Included | | - | |
| Industry F/RE(Dummy) | Included | | Included | | Included | |
| Year FE/RE(Dummy) | Included | | Included | | Included | |
| $R^2$ | 0.1777 | | 0.0709 | | 0.0523 | |
| Observations | | | 1178 | | | |

*Note1*: *, **, and *** indicate significance at the 10%, 5%, and 1% levels, respectively. *Note2*: One Religion is defined as one if the religion of a board is concentrated on one religion, and zero otherwise. Definitions of other variables are the same as the measurement of variables in Equation (1).

Rather, one religion board possibly exists only in a religious entity or group. Those firms likely internalize the ideology or philosophy of the religion, and so pursue social responsibility activities rather than seeking profitability and efficiency for profit maximization. Therefore, these results provide implications that, unlike other companies, firms with ownership structures related to religion may have other management purposes rather than maximizing profits for firm value.

## 5. Conclusions

This study examines whether the religious diversity of the board affects the firm value. BOD plays a critical role in the business through its monitoring and advising to the management. Hence, how well the board does its role is significantly crucial to firms. Of several topics related to the board composition for improving board effectiveness, the board diversity issue has recently received considerable attention. Prior studies examine various board diversity dimensions such as gender, age, race, ethnicity, and educational background and find evidence that board diversity positively affects the firm value. However, there is little research to examine the impact of the religious diversity of the board, due to perhaps data availability. Religion does affect the views, beliefs, and attitudes of individuals who belong to the religious group, and each religious group has its own value and belief distinct from other religious groups. Hence, the religious diversity of the board likely makes the board function well in a similar way that other dimensions of the board diversity do. Therefore, this study focuses on the religious diversity of directors and attempts to analyze how it affects firm value.

The analysis result of this study can be summarized as follows. First, the higher the level of religious diversity among board members, the higher the firm value. This means

that when the board consists of directors with diverse backgrounds in terms of religion, it properly monitors or advises the management in a way to increase management efficiency and profitability. Second, religious diversity and firm value have a non-linear relationship; when the level of religious diversity reaches a certain point, the relationship has the reverse U shape. In other words, the board religious diversity does not unconditionally increase firm value, but rather can negatively affect firm value after the religious diversity level exceeds the optimal level of diversity. Third, the firm value is relatively low when all board members have the same religion. This finding shows the possibility that firms owned by religious entity do a lot of activities or social contributions to realize religious ideology or value rather than increase the firm value.

This paper contributes to literature, expanding the dimension of board diversity. While diversity becomes one at the top of the corporate agenda and researchers have studied various dimensions of diversity, academia still does not pay attention to religious diversity due to data availability. At this point, this paper has implications by showing the impact of the religious diversity on firm value. In addition, findings of the low firm value with one-religion board give implications to the stock market participants.

However, this paper also has limitations as follows. Sample of this paper includes only KOSDAQ-listed firms in Korea that have two stock markets: KOSDAQ market and KOSPI market. Thus, our sample cannot represent all listed companies in Korea. Moreover, compared to the KOSPI-listed firms, these firms have small size and more volatile performance and stock returns. Hence, readers should cautiously interpret the results of this paper. In Korea, there is no one single dominant religion, but several religious groups co-exist. So, possibly this empirical result is not observed in other countries with very dominant a religious group, such as middle east countries.

**Author Contributions:** All authors contributed equally to this paper. Conceptualization, S.Y. and W.-Y.K.; methodology, S.Y.; formal analysis, W.-Y.K.; investigation, W.-Y.K.; resources, S.Y.; writing— original draft preparation, S.Y. and W.-Y.K.; writing—review and editing, S.Y., supervision, S.Y. and W.-Y.K. All authors have read and agreed to the published version of the manuscript.

**Funding:** This research received no external funding.

**Data Availability Statement:** Financial data will be available in KISVALUE (KISVALUE).

**Conflicts of Interest:** The authors declare no conflict of interest.

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
