# Peer review of "Religious Diversity of Corporate Board and Firm Value: Evidence from South Korea"

_religions, doi:10.3390/rel13050414_

Round 1
Reviewer 1 Report
A very interesting topic, unfortunately it is based on a limited number of cases and for this reason the results may be subject to variations when the available data will be greater. I read the text with great interest, thank you.
Author Response
A very interesting topic, unfortunately it is based on a limited number of cases and for this reason the results may be subject to variations when the available data will be greater. I read the text with great interest, thank you.

Reviewer 2 Report
Please check PDF file. Thank you!

Author Response
The study presents a detailed analysis of how religious diversity can be a growing value in the economic and human resources results of companies. Although the sample is not fully representative, it contributes to opening up a research horizon to be taken into account in the future. The work is excellent and with abundant and well-worked bibliography. The conclusions do not appear absolutized but rather in a demarcated business context, which means qualifying the results of the research.

Reviewer 3 Report
The paper engages with an interesting topic by discussing the relation between board religious diversity and firm value. The formulation of the main research hypothesis is clear enough. The author states that the main aim of the paper is to investigate the relation between religious diversity of the board and firm value in Korea from 2008 and 2011.
However there are many problems with the paper, both conceptually and methodologically. The conceptual analysis in the first part of the paper (2. Literature Review and Research Question) is not really adequate for the purpose of formulating a compelling research question. The paper is generally too short and the bibliography is not sufficient enough. Furthermore, the conceptual analysis is not detailed enough in order to achieve its main goals. The author argues that “religious diversity of the board has not received attention from practice and academia. There may be reasons for little attention to religious diversity” (lines 132-33). However, the author does not state these reasons. What is really missing in the Literature Review section of the paper is a thorough and in-depth conceptual analysis for the hypothesis that “diversity of the board is likely to affect the firm value positively” (lines 156-57). This remains an unclear hypothesis without adequate conceptual evidence or adequate argument formulation.
The next part of the paper (3. Research Methodology and Data) is related to the methodology and the data sample. However, there are also many problems with the methodology and the empirical evidence of the paper. The data sample is very limited, since it concerns, as the author herself/himself admits, only the KOSDAQ Association (Korea Securities Dealers Automated Quotation) (lines 166-74, 311-13). However, the main problem with the research methodology followed in this paper is that the data sample is no longer relevant, since it is already almost fifteen years old (2008 – 2011). It is really very difficult, if not impossible, to achieve reliable results, get substantive findings and formulate judgments with theoretical and empirical significance with such an old data sample. As a conclusion, it is impossible for me to formulate a positive judgment for this paper.
Author Response
Thank you very much for your thoughtful comments and constructive suggestions on our original version of the manuscript submitted to RELIGIONS. We believe that our paper has been improved after incorporating your valuable comments. In the following response, we present your original comments in bold italics, followed by our responses.
Point 1: There are many problems with the paper, both conceptually and methodologically. The conceptual analysis in the first part of the paper (2. Literature Review and Research Question) is not really adequate for the purpose of formulating a compelling research question. The paper is generally too short and the bibliography is not sufficient enough. Furthermore, the conceptual analysis is not detailed enough in order to achieve its main goals. The author argues that “religious diversity of the board has not received attention from practice and academia. There may be reasons for little attention to religious diversity” (lines 132-33). However, the author does not state these reasons. What is really missing in the Literature Review section of the paper is a thorough and in-depth conceptual analysis for the hypothesis that “diversity of the board is likely to affect the firm value positively” (lines 156-57). This remains an unclear hypothesis without adequate conceptual evidence or adequate argument formulation.
Response 1:
(1) Related to your comment that this paper lack of conceptual analysis, our response is as follows.
We understand your concern on lack of conceptual analysis in our paper. However, this paper is a first step of research projects connecting religion to corporation. We believe that more sophisticated conceptual analysis could be done in future projects.
In addition, this is a empirical research in economics field. We logically develop our hypothesis, based on prior literature which empirically examines about religion, board, and firm value. Our logic can be summarized as follows.
(a) Religion of corporate individuals affects corporate decision making by the corporate individuals. (b) Each religious group has different view and ideas, and so diverse background of board members in terms of the religion can bring diverse view and ideas as other diversity (e.g., gender, culture, expertise) does. (c) Board diversity (i.e., diversity with respect to composition of board members) enhances quality of monitoring and advising by the board. (d) High quality of monitoring and advising role by the board increase firm value. (e) Based on the logic above, we conjecture that the religious diversity of the board positively affect firm value.
(a), (b), (c), (d) are all empirically supported by prior literature. [2.1. Literature on the Diversity of Board] and [2.2. Literature on the Religion] includes prior literature of (a), (b), (c), (d).
(2) Related to your comment that this paper does not state reasons for academia’s little attention to religious diversity, our response is as follows.
In our original version of the manuscript, the following paragraph of the sentence “There may be reasons for little attention to religious diversity” provides possible reasons.
In the second paragraph of [2.3 Hypothesis Development], we suggest two reasons. First, religious diversity possibly has similar implication with cultural diversity. So, separating the religion effect and the cultural effect in terms of board diversity is less likely to have additional implications. La Porta et al. (1999), one of seminal paper in the field of corporate governance, indeed use religion as a proxy for culture in their study. Since this paper is about corporate board and firm value which is a main area of researchers in economics, finance, and accounting, we try to find the reason from a perspective of researchers in those areas. Second, religious groups may have a similar behavioral norm. In the section of [2.2 Literature on the Religion], we show that most of prior studies assume similar behavioral norm among individuals even in different religious groups and do not distinguish religious groups in their analysis. Under their assumption, religious diversity does not work as other dimensions’ diversity (e.g., gender diversity, cultural diversity) does since individuals even in different religious groups have similar view and behavioral norm. Under this argument, it would be more interesting to focus on the degree of religiosity rather than the religious diversity.
But, the sentence “There may be reasons for little attention to religious diversity” could mislead you to think that the reasons of little attention are not stated in this paper. So, we remove this sentence and include the following sentence.
[Line 141] in the revised version of the manuscript (writing in red is added).
We conjecture possible reasons for little attention as follows.
(3) Related to your comment that this paper does not provide conceptual analysis for the hypothesis that “diversity of the board is likely to affect the firm value positively”, our response is as follows.
In our original version of the manuscript, the paragraph that includes the sentence “Therefore, the religious diversity of the board is likely to affect the firm value positively” does provide hypothesis development about positive impact of the religious diversity. In that paragraph, we suggest the religious diversity of the board can enhance quality of advice by providing diverse view, idea, and information. Also, we suggest that monitoring quality by the board with greater religious diversity can be high since such religious diversity prevents the board from being a homogeneous group and avoids unchecked decision-making. There are many empirical papers showing that high quality of advice and monitoring by the board increase firm value. Therefore, we conjecture the positive relation between the religious diversity and firm value.
Moreover, in the section of [2.1. Literature on the Diversity of Board], many prior studies in economics, finance, and accounting research areas provide empirical evidence that high quality of monitoring and advising role by the board increase firm value. In addition, in the second paragraph of [2.2. Literature on the Religion], prior papers empirically show that the religion of corporate decision makers do affect corporate behavior. Also, in the third paragraph of [2.2. Literature on the Religion], some of empirical studies show each religious group has different view on several issues, which suggest the religious diversity can work as a mechanism for incorporating diverse view and idea.
Based on the above argument provided in the sections of [2.1. Literature on the Diversity of Board] and [2.2. Literature on the Religion], we develop the hypothesis of the positive relation between firm value and the religious diversity as written in the third paragraph of the section [2.3. Hypothesis Development: Relationship between Religious Diversity and Firm Value].
Point 2: The next part of the paper (3. Research Methodology and Data) is related to the methodology and the data sample. However, there are also many problems with the methodology and the empirical evidence of the paper. The data sample is very limited, since it concerns, as the author herself/himself admits, only the KOSDAQ Association (Korea Securities Dealers Automated Quotation) (lines 166-74, 311-13). However, the main problem with the research methodology followed in this paper is that the data sample is no longer relevant, since it is already almost fifteen years old (2008 – 2011). It is really very difficult, if not impossible, to achieve reliable results, get substantive findings and formulate judgments with theoretical and empirical significance with such an old data sample. As a conclusion, it is impossible for me to formulate a positive judgment for this paper.
Response 2:
As you pointed out, our sample includes only companies listed in KOSDAQ market. Korea has two stock markets: KOSDAQ and KOSPI. So, our sample cannot represent all listed firms. But, in Korea, Korea stock market disclosure rule does not require companies to disclose directors’ religion while it requires disclosure of gender, age, education, previous career of directors on the board. Companies do not even voluntarily disclose directors’ religion because it is very private information. Not just Korea but also other countries do not mandate disclosure of directors’ religion. So, due to data availability, prior studies have not empirically analyzed the relation between firm value and religion-related variables.
However, KOSDAQ association did survey for various information of individual directors in all KOSDAQ-listed companies from 2008 to 2011. In that survey, director’s religion information was also collected. After that, KOSDAQ association stopped doing the survey. So, data of directors’ religion is available only for 2008-2011. That is the reason we use limited and old sample in our empirical analysis.
We admit that our sample is limited and old. But, as aforementioned, data of directors’ religion is very unique and rare. So, we think that this data is worth utilizing to figure out the relation between firm value and religious diversity that has not been studied due to data availability.
But, we stated this limitation in the fourth paragraph of [5. Conclusion] in our original version of the manuscript. But, to incorporate your comment, we revised our manuscript as follows. (writing in red is added)
Line 175-184 in the revised version of the manuscript [3. Research Methodology and Data]
Data on the religion of corporate BOD members are hard to obtain since there is no disclosure rule mandating corporate disclosure of directors’ religion. Furthermore, com-panies do not even voluntarily disclose directors’ religion since it is private information. Thus, due to data availability, studies on the relationship between religious factors and firm values have not been actively examined so far. However, Korea Securities Dealers Automated Quotation(hereafter, KOSDAQ) Association conducts a survey for four years from 2008 to 2011 on directors’ religion of manufacturing companies listed in the KOSDAQ stock market. We obtain the data of directors’ religion from KOSDAQ Association and so limit our sample to KOSDAQ-listed companies from 2008 to 2011. Admitting that our sample is limited sample, we utilize this data to examine our research question because this data is unique and rare.
Line 373-376 in the revised version of the manuscript [5. Conclusions]
However, this paper also has limitations as follows. Sample of this paper includes only KOSDAQ-listed firms in Korea that have two stock markets: KOSDAQ market and KOSPI market. Thus, our sample cannot represent all listed companies in Korea. Moreover, compared to the KOSPI-listed firms, these firms have small size and more volatile perfor-mance and stock returns.

Round 2
Reviewer 3 Report
The manuscript has been sufficiently improved and is suitable for publication.